Testing the ‘parasite-mediated domestication’ hypothesis: a comparative approach to the wild boar and domestic pig as model species

Oleinic Renat 1
Posedi Janez 2
Beck Relja 3
Šprem Nikica 4
Škorput Dubravko 5
Pokorny Boštjan 6 7
Škorjanc Dejan 1
Prevolnik Povše Maja 1
Skok Janko janko.skok@um.si 1
1 Faculty of Agriculture and Life Sciences, Department of Animal Science, University of Maribor , Maribor , Slovenia
2 Veterinary Faculty, Institute of Microbiology and Parasitology, Unit for Parasitology, University of Ljubljana , Ljubljana , Slovenia
3 Laboratory for Parasitology, Croatian Veterinary Institute , Zagreb , Croatia
4 Faculty of Agriculture, Department of Fisheries, Apiculture, Wildlife Management and Special Zoology, University Zagreb , Zagreb , Croatia
5 Faculty of Agriculture, Division of Animal Science, University Zagreb , Zagreb , Croatia
6 Faculty of Environmental Protection , Velenje , Slovenia
7 Slovenian Forestry Institute , Ljubljana , Slovenia
Gelfand Mikhail
Electronic publication date: 2024 Nov 25
Publication date: 2024
Volume: 12
Electronic Location ID: e18463
Received 2024 May 16; Accepted 2024 Oct 15
Copyright: ©2024 Oleinic et al.
Copyright year: 2024
Copyright holder: Oleinic et al.
License: This is an open access article distributed under the terms of the Creative Commons Attribution License, which permits unrestricted use, distribution, reproduction and adaptation in any medium and for any purpose provided that it is properly attributed. For attribution, the original author(s), title, publication source (PeerJ) and either DOI or URL of the article must be cited.
License URL: https://creativecommons.org/licenses/by/4.0/

Keywords: Domestication, Domestication syndrome, Endoparasites, Helminths, Protozoa, Sus scrofa

Funding: The Slovenian Research and Innovation Agency P1-0164 P4-0092 P4-0107 N4-0350 Ministry of Agriculture, Forestry and Food of Republic of Slovenia project V4-2223 Croatian Science Foundation HRZZ IP-2022-10-7502 The study was supported by the Slovenian Research and Innovation Agency (research programmes No. P1-0164, P4-0092, P4-0107; project N4-0350), Ministry of Agriculture, Forestry and Food of Republic of Slovenia (project V4-2223), and the Croatian Science Foundation HRZZ (IP-2022-10-7502). The funders had no role in study design, data collection and analysis, decision to publish, or preparation of the manuscript.

==============================
The mechanisms underlying the domestication process have already been well explained. Starting with Belyaev’s pioneering experiment on silver foxes, which showed that selection for tameness (reduced fear response, changes in the ‘hypothalamic-pituitary-adrenal system’—HPA axis) leads to destabilisation of the regulatory systems that control morphological and behavioural development, resulting in the changes characteristic of the domestication syndrome. Later, the thyroid rhythm hypothesis and the neural crest cell (NCC) hypothesis provided additional explanations. Recently, the parasite-mediated domestication hypothesis (PMD) has been proposed, suggesting an important role of endoparasites in the domestication process. Since parasites either directly or indirectly affect all mechanisms underlying the domestication syndrome (HPA, thyroid, NCC), the PMD proposes that they may be an important trigger of domestication. PMD can be tested either experimentally or comparatively. One of the basic assumptions of PMD is that parasite-susceptible, genetically less resistant/more tolerant wild animals were originally domesticated and that the susceptibility to parasites has been passed on to today’s domestic animals. This can be verified primarily by comparing the parasite susceptibility of existing wild and domestic populations of the same species. We, therefore, followed a systematic comparative approach by analysing the parasite load in wild boar (WB) and free-ranging domestic pig (DP) populations from a comparable environment in a geographically close area. Fresh faeces from WB and DP populations, one each from Slovenia (SI) and Croatia (HR), were sampled. A total of 59 individual faecal samples were collected (SI: 12 WB, 20 DP; HR: 14 WB, 13 DP). Parasitological diagnostics were carried out using the sedimentation and floatation method. Five different taxa were found in WB and seven in DP. Three parasite taxa were found exclusively in DP (Cystoisospora suis, Trichuris sp., Balantidium coli), and one (Strongyloides sp.) only in WB. Of the parasites found in both cohorts, strongyles/Oesophagostomum sp. were significantly more abundant in DP, while Eimeria sp. was found in equal amounts in both (but in HR only in WB). According to the preliminary study presented here, there is evidence to support the PMD baselines in the wild boar—domestic pig association. However, we cannot draw a definitive conclusion as there are many aspects that may bias the interpretation based on parasite load alone, which are also discussed here. Therefore, comparative studies should be supported by a more focussed methodology, including an experimental approach.

Introduction

Domesticated animals are characterised by behavioural and morphological traits that are referred to as domestication syndrome. Among domestication phenotypic traits, tameness was an essential prerequisite, followed by floppy ears, a short and curled tail, piebaldism, depigmentation, short and wide skull, reduced size of the adrenal gland, and others (Belyaev, 1979; Wilkins, 2017). Many of the mechanisms behind the process of domestication and thus the domestication syndrome have already been well explained. Starting with the pioneering studies on domestication by Belyaev (1979), who proposed that the domestication syndrome is genetically linked to genes associated with tameness. His experiment on domestication of silver foxes, which is among the most influential work in this field, showed that selection for tameness (reduced fear response, changes in the ‘hypothalamic-pituitary-adrenal system’—HPA axis) leads to significant destabilisation of regulatory systems controlling morphological and behavioural development, resulting in changes that are otherwise characteristic of domestication syndrome. Much later, some other findings and hypotheses were added (Wilkins, 2017). In the thyroid rhythm hypothesis, Crockford (2004) proposed that domestication is also driven by genetically controlled changes in the activity rhythm of the thyroid gland, which have a crucial effect on heterochronic changes and thus play an important role in the domestication syndrome (e.g., paedomorphism). Furthermore, Wilkins, Wrangham & Fitch (2014) proposed that the main phenotypic components of domestication syndrome are neural crest cells (NCC) derivatives, i.e., a result of a developmental reduction in NCC input for the affected phenotypic traits. Recently, the parasite-mediated domestication hypothesis (PMD) has been proposed (Skok, 2023). PMD derives from the fact that parasites literally affect all the major mechanisms that otherwise underlie the domestication syndrome (HPA, thyroid, NCC) and in this way could mediate the domestication process, both directly through the manipulation of regulatory systems and behavioural traits of the host (Poulin, 1994; Poulin, 2013; Adamo, 2013) and indirectly through genes related to resistance/tolerance to parasites, the role of circulating miRNAs in the process of epigenetic inheritance or epigenetic transgenerational inheritance of stress pathology (Sharma, 2014; Van Otterdijk & Michels, 2016; Matthews & Phillips, 2012). Ultimately, domestication syndrome traits may be genetically linked to genes related to resistance and tolerance to parasites, with (proto)domestication selection favouring less genetically resistant (more tolerant) individuals in which domestication syndrome traits are significantly more common due to the influence of parasites, according to the PMD (Skok, 2023).

There are several mechanisms that could hypothetically be involved in the mediation of domestication by parasites and thus in the triggering of the domestication phenotype. First, parasites often trigger stress pathologies and influence host behaviour in a way that increases their transmission probability (Poulin, 2010; Poulin, 2013; Del Giudice, 2019; Hughes & Libersat, 2019)—in particular, they may reduce fear responses to increase inter- and intraspecific contacts. Indeed, in the early stages of domestication, reduced animal fear of humans (i.e., tameness) was necessary for life in an anthropogenic environment (Herbeck et al., 2022). Second, parasites can significantly affect the profile of miRNAs, either those affecting (neuro)endocrine function, including hormone production and concentration, or those significantly involved in the migration and development of NCCs that shape the phenotype of domestication, e.g., miRNAs involved in chondrogenesis and bone development, craniofacial development, pigmentation, etc. (reviewed by Skok, 2023). In addition, parasites often cause disruptive selection and thus increased genetic and phenotypic variability within the host population (Duffy et al., 2008; Blanchet et al., 2009), an otherwise well-known phenomenon in domesticated populations.

Therefore, the PMD assumes an important role of endoparasites in the process of domestication, especially in the initial phase (proto-domestication). It predicts that the frequency of domestication syndrome traits in the wild population increases with decreasing genetic resistance/increasing tolerance to parasites and with increasing parasite load (Skok, 2023). Thus, there are two main PMD baselines that can be tested in different ways: (i) the proposition that parasite-susceptible, genetically less resistant wild animals were originally domesticated and that susceptibility to parasites has been passed on to today’s domestic animals can be tested by comparing parasite load (or genetic resistance/tolerance) in existing wild populations (expected lower load/higher resistance) and domesticated populations (expected higher load/lower resistance) of the same species; and (ii) the occurrence of domestication syndrome traits is predicted to be significantly higher in highly infected wild populations, which can be tested either comparatively (frequency of domestication syndrome traits in highly parasitised wild populations compared to less infected wild populations) or experimentally (frequency of domestication syndrome traits in experimentally parasitised wild populations).

However, testing PMD based on parasite load, resistance and tolerance in existing populations of wild or domestic animals and simply comparing these populations is interpretatively quite complex. Comparative studies should be supported by a more focussed methodology. Either to examine the frequency of domestication syndrome traits in the wild population in relation to their parasite resistance, tolerance and load, or to examine the parasite resistance, tolerance and load of wild animals showing signs of domestication syndrome (e.g., tameness) in comparison to completely wild animals of the same population, as originally proposed (Skok, 2023). The other and probably most reliable way to test PMD would be an experimental approach, i.e., experimental (proto)domestication, such as the Belyaev fox experiment (Belyaev, 1979) or the Trapezov mink experiment (Trapezov, 1987). However, instead of selecting animals for tameness, an experimental population of the wild counterpart of domestic animals would be experimentally exposed to parasites and selected for parasite resistance/tolerance. The frequency of domestication syndrome traits in the populations with varying degrees of parasite susceptibility would then be analysed over generations. It is predicted that the frequency of typical domestication syndrome traits (tameness, depigmentation, floppy ears, etc.) would increase with increasing susceptibility to parasites.

Although comparative studies can be problematic for a variety of reasons, they are relatively easy to perform and are, therefore, suitable for initial testing of hypothesised PMD baselines. According to the PMD baseline of an originally impaired genetic, and thus heritable, parasite resistance in wild individuals, which led to domestication, it can be assumed that the parasite load in the domestic population under comparable conditions is still higher today than in the wild population. Accordingly, we tested PMD in Sus scrofa using a systematic comparative approach by analysing the parasite load in wild boar and free-ranging domestic pigs from a comparable environment. We selected animals from two regions, one in Slovenia and the other in Croatia. In each region, samples were collected from wild and domestic cohort in a relatively narrow geographical area. According to PMD, we proposed a higher parasite load in the domestic pig compared to its wild counterpart, both in terms of diversity of parasite taxa and quantitatively (severity of infection).

Materials & Methods

Study sites and sampling period

Following the PMD predispositions, we aimed to compare the parasite load between domestic pigs and wild boar living in the same area. Two test areas in Croatia (HR) and Slovenia (SI), about 150 kilometers apart, were included in the study (Fig. 1). The samples in both countries consisted of two cohorts, wild boar and domestic pigs. In both countries, the sampling sites of the wild boar and domestic pig populations were about 15–20 km apart, so that all animals had similar chances of being exposed to the local parasitofauna. In Croatia, the wild boar samples were collected on 3 and 4 November 2023 in the Prolom hunting ground (village Buzeta, municipality of Glina, Sisak-Moslavina County, Croatia) and the domestic pig samples were collected on 10 November 2023 in the village Srednje Mokrice (municipality of Petrinja, Sisak-Moslavina County, Croatia) in the approximately 3-hectare outdoor (mainly grassland) enclosure of the autochthonous Banija spotted pig (Banijska šara). In Slovenia, the wild boar samples were collected on 10 December 2023 in the hunting ground of Stoperce (village Stoperce, municipality of Majšperk, Štajerska region, Slovenia) and the domestic pig samples were collected on 11 December 2023 in the village Zgornje Laže (municipality of Slovenjske Konjice, Štajerska region, Slovenia) in the approximately 3-hectare outdoor (mainly grassland) enclosure of the autochthonous Krškopolje pig (Krškopoljski prašič).

Figure 1 Map of the wild boar –domestic pig sites in both test areas.

Geographical location of the Slovenian test area (SI) and the Croatian test area (HR), each consisting of the two cohorts studied, wild boar (WB, green square with dot) and domestic pig (DP, red circle with dot). Map created with Google Earth Pro.

Both outdoor pigsties have been in operation for around 10–15 years, so the parasite presence could add up over the course of permanent inhabitation—but the same applies to wild boar, which have also been living permanently in relatively high densities in the hunting grounds for decades.

Animals and faeces collection

Wild boar populations and free-ranging (grazing) domestic pig populations, one each from Croatia (HR) and Slovenia (SI), were included in the study. Fresh faeces were collected from the rectum of hunted wild boar or immediately after the defecation of domestic pigs. Each individual sample was collected in 100 ml plastic cup with lid. Before taking faecal samples from each animal, the rubber gloves were changed to avoid cross-contamination of the individual samples. The samples were then labelled accordingly and stored in a refrigerator at a temperature of 4 °C and analysed up to 48 h after sampling. A total of 59 individual faecal samples (SI: 12 wild boar, 20 domestic pig; HR: 14 wild boar, 13 domestic pig) were examined for endoparasites. Domestic pigs were not treated against parasites.

The faecal samples were taken either from regularly shot wild boar in the hunting ground or from domestic pigs on the farm in the presence of representative persons. The hunting ground in Croatia is state-owned and is under the responsibility and administration of the University of Zagreb (authorisation No. 251-71-29-02/19-22-2).The hunting ground in Slovenia is state-owned and is under the responsibility and administration of the Stoperce Hunting Club (member of HAS, The Hunters Association of Slovenia). The faecal samples from domestic pigs were taken on the pig farm with the permission of the farm owners Bojan Lešnik (Slovenia) and Željko Horvat (Croatia).

Diagnosis of parasitic infections

Due to the occurrence of African swine fever (ASF) in Croatia (the first occurrence of the virus was confirmed in domestic pigs on June 26, 2023), which also hindered the sampling process in general, all samples remained in the country of origin and were analysed there. The Croatian samples were analysed in the Laboratory for Parasitology of the Croatian Veterinary Institute in Zagreb. Slovenian samples were analysed in the Unit for Parasitology of the Institute of Microbiology and Parasitology at the Veterinary Faculty of the University of Ljubljana.

The faecal samples were examined for the presence of endoparasites by flotation using two different quantification methods: the semi-quantitative (SI) or McMaster method, i.e., counting (HR), and the sedimentation method (semi-quantitative SI; counting HR) according to Thienpont, Rochette & Vanparijs (2003). A saturated salt solution (density 1.20 at 20 °C) was used to float nematode and cestode eggs/(oo)cysts, while tap water (max. 30 °C) was used for sedimentation. Taxon determination under the microscope was based on the different morphotypes of the eggs/(oo)cysts. The parasite load assessment differs between national laboratories, but this was irrelevant to the reliability of the PMD as we were primarily aiming to compare wild and domestic populations within the selected test area in each country, rather than between countries, as we were interested in comparing animals living in the same area and potentially encountering the same local parasitofauna. Comparing parasite loads between countries does not provide relevant information in relation to PMD, as the SI and HR populations were geographically too far apart and, therefore, may not have been exposed to comparable parasitofauna.

In its standard quantification methodology, the Slovenian laboratory uses five classes for the intensity of parasite load, defined by the number of eggs/(oo)cysts present in the sample/smear. The classes were defined as follows: Negative = 0; Individual egg/oocyte = 3-5; + = 5–10; ++ = 10–20; +++ ≥21. In the Croatian laboratory, the total number of eggs/(oo)cysts per gram in the sample of a given quantity (approx. 2-5 g) was defined. For this reason, the results are presented here separately for each country and with a different scale for the figures.

Data analyses

Due to the restriction with regard to ASF and the separate diagnostics within the country from which the samples originated, the estimates of the parasite load differed between the two data sets and were, therefore, also statistically analysed differently. For the Slovenian data, where the parasite load estimators were reported as classes, the non-parametric Mann–Whitney U-test (also known as Wilcoxon rank sum test) was used to compare the parasite load between the domestic and wild cohorts. The Croatian estimator was a continuous variable, however, as it was not normally distributed, the non-parametric Independent-Samples Wald-Wolfowitz Run test was used to compare the parasite load between the domestic and wild cohorts.

Results

As shown in Table 1, a total of five different parasite taxa were found in the HR samples. Wild and domestic populations were each infected with three different parasite taxa (wild boar: Eimeria sp., strongyle type eggs, and Strongyloides sp.; domestic pig: Cystoisospora suis, strongyle type eggs, and Trichuris sp.). A total of four different parasite taxa were found in SI samples. Only two of them were found in wild boar, Eimeria sp., and Oesophagostomum sp. (strongyle), but all four (Eimeria sp., Balantidium coli, Oesophagostomum sp. (strongyle), and Trichuris sp.) were found in domestic pig. Strongyloides sp. was the only parasite taxon found in wild boar but not in domestic pigs, all others were found either in domestic pigs exclusively or in both cohorts.

Table 1 Description of the parasites found in the study.

Parasite taxa found in the study with summarised taxa richness (S) for each cohort.

	Domestic pig	Wild boar	S	
HR	Cystoisospora suis a	Eimeria sp.	5	
	Strongyle type eggs	Strongyle type eggs		
	Trichuris sp.a	Strongyloides sp.b	
SI	Eimeria sp.	Eimeria sp.	4	
	Oesophagostomum sp. (strongyle)	Oesophagostomum sp. (strongyle)		
	Trichuris sp.a			
	Balantidium coli a			
S	7	5		
Notes.

a Only domestic cohort infected

b Only wild cohort infected

HR Croatia

SI Slovenia

In the HR samples, the only taxon present in both cohorts were strongyles, the eggs of which were significantly more abundant in domestic pigs than in wild boar. Of the others, Cystoisospora suis and Trichuris sp. were found exclusively in domestic pigs, while Eimeria sp. and Strongyloides sp. were only found in wild boar (Fig. 2).

Figure 2 Parasite load of the wild boar-domestic pig association in the HR test area.

Comparison of the intensity of infection with certain parasite taxa between wild boar and domestic pig cohorts in Croatia. The centre lines of the boxplots indicate the medians, the edges of the boxes represent the 25th and 75th percentiles. 1 indicates that only one cohort was infected, TS, test statistics; r, effect size.

In the SI samples there were two parasite taxa, Trichuris sp. and Balantidium coli, which were found exclusively in domestic pigs, while the other two were found in both cohorts. For two matching parasites, the load of Oesophagostomum sp. (strongyle) was significantly higher in domestic pig than in wild boar, while Eimeria sp. was present to the same extent in both cohorts (Fig. 3).

Figure 3 Parasite load of the wild boar-domestic pig association in the SI test area.

Comparison of the intensity of infection with certain parasite taxa between wild boar and domestic pig cohorts in Slovenia. The centre lines of the boxplots indicate the medians, the edges of the boxes represent the 25th and 75th percentiles. In the figure, 1 indicates that only one cohort was infected, U, test statistics; r, effect size.

Discussion

Domestic pigs generally had a higher parasite load than their wild counterparts. The results of the present study, therefore, support one of the basic assumptions of PMD that domestic pigs could descend from parasite-susceptible wild ancestors, whereby the susceptibility to parasites, which presumably trigger the domestication syndrome, is then passed on/inherited through generations of progeny during the domestication process. Rare previous comparative studies on parasite load in Sus scrofa have shown mixed results, some of which contradict ours—although it should be emphasised that the sampling in previous comparative studies was not as targeted as ours. Ineson (1954), for example, found differences in the parasite load depending on the taxon of the parasite, with domestic pigs having a higher overall parasite load (richness) than wild boar. In contrast, Allwin et al. (2015) found a higher parasite load in wild boar than in domestic pigs in semi free-range or on farms.

Indeed, based on the results, we cannot draw a definitive conclusion, because apart from the fact that this is only a very preliminary attempt at a systematic comparison to test PMD, there are many aspects that could confound interpretation based on parasite load alone.

Genetic resistance/tolerance to parasites associated with PMD tested by comparison of parasite load

The parasite load depends strongly on the genetic resistance and tolerance of the animals to parasites (Råberg, Graham & Read, 2009). In this context, the first aspect concerns the possible artificial selection in domestic animals aimed at increasing parasite resistance/tolerance (McManus et al., 2014), possibly leading to a deceptively high resistance/tolerance to parasites in the domestic population. However, this was not the case in our study, as none of the breeds included in our study had undergone methodical selection for parasite resistance and tolerance in the past, although they may have been passively selected. In addition, selection for parasite resistance/tolerance in pigs has generally not received much attention, although the heritability of host resistance/tolerance to certain parasites is so high that breeding for resistant pigs may be a possibility (Roepstorff et al., 2011).

The second aspect, which can lead to the opposite conclusion, concerns relaxed selection, in which the source of selection that was previously important for the maintenance of a particular trait is weakened or even eliminated (Lahti et al., 2009). Indeed, selection pressure from parasites may have been reduced with domestication, as the animals were confined, at least to some degree, so they may not have been exposed to the full range of parasites in their natural environment. This, together with “stringent” artificial selection for desirable traits, could influence (jeopardise) the evolution of genetic resistance and tolerance in (parasite-naïve) domestic animals (Eizaguirre et al., 2012; Smallbone et al., 2021).

Regarding genetic parasite resistance and tolerance, the major histocompatibility complex (MHC) is often considered one of the functionally most important loci associated with parasite resistance/tolerance (Sommer, 2005; Axtner & Sommer, 2012; Arbanasić et al., 2019). It is assumed that animals with limited MHC gene diversity also have low resistance/tolerance to parasites. Indeed, constant selection pressure by various parasites leads to rapid adaptive evolutionary changes implying that parasite-mediated selection directly contributes to the maintenance of MHC polymorphism (Eizaguirre et al., 2012). The lower allelic diversity in genes associated with parasite resistance and tolerance could, therefore, also be due to a bottleneck/inbreeding effect or a generally smaller gene pool due to domestication. While the bottleneck effect appears to influence parasite resistance in domesticated guppies (Smallbone et al., 2021) and the red junglefowl (Nguyen-Phuc, Fulton & Berres, 2016), studies in pigs did not follow the same premise and showed that domestication does not appear to act as a bottleneck limiting MHC diversity (Moutou et al., 2013); and the same was found in the zebra finch (Newhouse & Balakrishnan, 2015). However, MHC diversity does not necessarily imply higher functional resistance and tolerance but may simply be the signature of the pathogens and parasites that the animal has been confronted with during evolution (Mikko et al., 1999; Portanier et al., 2019, showing that MHC diversity in mouflons is associated with resistance to nematodes but not to coccidia). In addition to MHC, there are other quantitative trait loci for parasite resistance and tolerance (McManus et al., 2014), which makes this aspect of PMD testing even more challenging.

The possible role of certain parasites only in PMD

The influence of parasites on the domestication process cannot simply be generalised, as it is also possible that only certain parasites are involved in mediating the domestication process.

To mediate domestication, direct interspecific transmission (i.e., without an intermediate host) including a possible zoonotic potential of the parasite seems to be a necessary criterion. Of the parasite taxa found in our study, all parasites found exclusively or predominantly in domestic pig (Balantidium coli, Strongyle/Oesophagostomum sp. and Trichuris sp.) fulfil both criteria, but not Cystoisospora suis, which was only found in domestic pig and is species-specific. In addition, the only parasite taxa found exclusively in wild boar was Strongyloides sp. of which none of the known species infecting pigs were found to be zoonotic but have been shown to be both parthenogenetic and autoinfectious (Moore, 2002), making them less reliant on intensive transmission between hosts. Further, the PMD predicts that parasites should increase the occurrence of domestication syndrome traits. For the parasites found in our study, there is a significant lack of specific information on their potential effects on phenotypic traits of interest in the context of PMD. However, of the target traits, mainly behavioural changes/correlates of the host have been investigated so far. Studies on the impact of the endoparasites taxa relevant (found) in our study on host behaviour usually deal with either parasite load in relation to a specific personality type or social behaviour (Côté & Poulin, 1995; Ezenwa, 2004; Melfi & Poyser, 2007; Müller-Klein et al., 2019; Santicchia et al., 2019), avoidance/sickness behaviour within a social group (Ghai et al., 2015; Chapman et al., 2016; Friant, Ziegler & Goldberg, 2016; Wren et al., 2021), changes in the complexity and dynamics of a social group (Burgunder et al., 2017) and the fear response to predators (Moore, 2002; Barber & Dingemanse, 2010; Kaushik, Lamberton & Webster, 2012). Indeed, behaviours that are (in)directly indicative of the animals’ degree of tameness, which is otherwise a pivotal trait of the domestication syndrome, are primarily of PMD interest.

Tameness in the context of domestication can be defined as reduced fear of humans, as well as increased boldness and reduced aggression, which have been shown to cause the development of many aspects of the domestication syndrome (Belyaev, 1979; Agnvall et al., 2015). With regard to PMD, the influence of parasites on behaviours such as reduced fear of novelty or of other species, including humans is, therefore, of particular interest.

Of the parasites found in the present study, some (Eimeria sp., Strongyles, Strongyloides sp., Trichuris sp.) were associated with host behaviour, but the results are not conclusive. Eimeria sp., for example, has generally been shown to reduce fear of predators (Moore, 2002; Kaushik, Lamberton & Webster, 2012). However, when looking at neophobia/boldness, for example, it was found that rural and urban rats, the latter being considered less neophobic (bolder), did not differ in Eimeria sp. load, but were more infected with Trichuris sp. (Battersby, Parsons & Webster, 2002). In addition, bolder and more exploratory individuals of grey squirrels were shown to be more heavily infected with Strongyloides sp. (S. robustus), but this was considered to be a cause rather than a consequence of infection severity (Santicchia et al., 2019). Given the apparent species specificity, potential studies on our model species (Sus scrofa) are of particular importance, but they are sparse and indirect. In wild boar, a study was recently conducted on parasite load in relation to the degree of urbanisation of the area in which they forage. No general difference in endoparasite load (including one strongyle species, Oesophagostomum dentatum) was found between wild boar foraging in suburban areas and those foraging in urban areas—except for Eimeria sp., the prevalence of which was significantly higher in individuals foraging in urban areas, i.e., those that were less afraid of the anthropogenic environment/humans (Pilarczyk et al., 2024).

Although not of paramount importance in the context of PMD, intraspecific interactions among members of the same social group have also been frequently studied and should, therefore, be considered. Indeed, literally all domesticated animals except the cat are social, with the dominant status of an individual indirectly indicating the degree of tameness. In general, subordinate animals are considered less aggressive/tameable (Blanchard et al., 1988; Holekamp & Strauss, 2016) and (with some exceptions) less neophobic, i.e., they are bolder and tend to explore new environments and objects (King, 1973; Robertson, 1982; Johnson & Balph, 1990; Darrow & Shivik, 2009; Mettler & Shivik, 2007; Schaffer et al., 2021). Among the parasites found in our study, susceptibility to infection with some of them was also found to be related to individual social status (dominant vs. subordinate). In plain zebras, for example, it was found that subordinate individuals, which could also be considered tamer, were more susceptible to infection with strongyles as well as Strongyloides sp. (Fugazzola & Stancampiano, 2012; Joly et al., 2023). In contrast, in bovids (several species of gazelles), territorial males, which can be considered dominant, have been found to have a higher strongyle and coccidia (including Eimeria sp.) load (Ezenwa, 2004).

However, our results do not fit easily into the picture of previous findings on the relationship between specific parasite and tameness, as we found Eimeria sp. to the same extent in the domestic and wild cohort; moreover, in the Croatian sample it was found only in the wild cohort, while strongyle/Oesophagostomum sp. and Trichuris sp. were found in domestic pig to a significantly higher extent or exclusively in the domestic cohort, and Strongyloides sp. was found only in wild boar.

Conclusions

The main findings of the present study are consistent with the PMD in terms of overall parasite load/richness, as more parasite taxa were found in the domestic pig. Of the parasites found in both cohorts, the infection with strongyle/Oesophagostomum sp. was significantly higher in the domestic pig, but not with Eimeria sp. which was found to the same extent in both cohorts. However, as already mentioned, such a comparative study is not conclusive, so that clear answers can only be obtained by analysing the frequency of domestication syndrome traits in wild populations as a function of the degree of parasite infection or by setting up experimentally parasitised wild populations that are examined for the frequency of domestication syndrome traits. Also, when considering the influence of a specific parasite on domestication syndrome traits, no clear conclusions can be drawn as the relevant studies on the influence of parasites on phenotypic traits are limited and some of the parasites found have not been studied at all in this respect. In addition, the studies to date suggest that the influence of a particular parasite may even be species-specific, but the studies on pig (Sus scrofa) are sparse (and only indirect). Therefore, we can only derive partial and indirect correlations that provide some indices in favour of PMD, but no definitive answers. This was neither possible nor expected given the nature of the study, as it was a preliminary test of the hypothesised baselines of PMD.

Our special thanks go to the domestic pig breeders Bojan Lešnik (Slovenia) and Željko Horvat (Croatia), who made it possible for us to obtain samples from domestic pigs, and to Luka and Boštjan Žunkovič, who enabled access to samples from wild boar in Slovenia.

Additional Information and Declarations

Competing Interests

Author Contributions

Animal Ethics

Field Study Permissions

Data Availability

The authors declare there are no competing interests.

Renat Oleinic conceived and designed the experiments, performed the experiments, analyzed the data, prepared figures and/or tables, authored or reviewed drafts of the article, and approved the final draft.

Janez Posedi conceived and designed the experiments, performed the experiments, analyzed the data, authored or reviewed drafts of the article, and approved the final draft.

Relja Beck conceived and designed the experiments, performed the experiments, analyzed the data, authored or reviewed drafts of the article, and approved the final draft.

Nikica Šprem conceived and designed the experiments, performed the experiments, authored or reviewed drafts of the article, and approved the final draft.

Dubravko Škorput conceived and designed the experiments, performed the experiments, authored or reviewed drafts of the article, and approved the final draft.

Boštjan Pokorny conceived and designed the experiments, authored or reviewed drafts of the article, and approved the final draft.

Dejan Škorjanc conceived and designed the experiments, authored or reviewed drafts of the article, and approved the final draft.

Maja Prevolnik Povše conceived and designed the experiments, analyzed the data, prepared figures and/or tables, authored or reviewed drafts of the article, and approved the final draft.

Janko Skok conceptualised the research, conceived and designed the experiments, performed the experiments, analyzed the data, prepared figures and/or tables, authored or reviewed drafts of the article, and approved the final draft.

The following information was supplied relating to ethical approvals (i.e., approving body and any reference numbers):

The faecal samples were taken either from regularly shot wild boar in the hunting ground or from domestic pigs on the farm immediately after defecation. Therefore, the authorisation of the competent authority was not required for our work.

The following information was supplied relating to field study approvals (i.e., approving body and any reference numbers):

The faecal samples from domestic pigs were taken on the pig farm with the permission of the farm owner.

The following information was supplied regarding data availability:

The raw data of parasite diagnostics are available in the Supplemental File.

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
