# Peer review of "Testing the ‘parasite-mediated domestication’ hypothesis: a comparative approach to the wild boar and domestic pig as model species"

_PeerJ, doi:10.7717/peerj.18463_

## Round 0.1 · original submission · Major Revisions

While both authors consider the manuscript to be potentially interesting, they suggest a number of editorial comments ranging from minor editing corrections to more substantial rewriting. Of the latter, the most important one is that there is a need to more carefully explain the mechanisms of the presumed influence of parasite load on traits emerging during domestication, and how one could separate this influence from other, more obvious contributions, e.g. that of the hormonal status (the explanation set up in the pioneering works of Belyaev: selection for less aggression selects for less testosterone, hence side effects).

Reviewer 1 ·

Basic reporting

The authors present an intriguing hypothesis and a general approach to test it. I see issues in the lack of explanation in regards to the mechanistic aspects of the suggested hypothesis. The way it is currently presented it doesn’t really seem to explain the domestication syndrome, but rather something that happens alongside domestication due to bottlenecking/inbreeding or general smaller gene pool. I as the reader would need more details on how the hypothesis works to really understand what the authors claim the parasites contribute in the domestication process. Overall the introduction is missing references and the methodology is lacking details and varies between sites.

Experimental design

The authors are upfront with the potential downfalls of their approach and mention them in the abstract. There are predictions to the hypothesis, but the hypothesis is ill explained and I am not able to make the connections that are implied. Since this is a new hypothesis that lays the foundation for the paper the authors should explain the mechanistic workings of said hypothesis in more detail to help the reader understand the relevance, experimental design, and connections. The experimental design is lacking details in the methodology described and applies different approaches to the two study sites used.

Validity of the findings

This paper presents an intriguing idea. With the minimal information provided in the methods I’m not sure if we can really draw any conclusions from the results. See my comments below on the enclosure setup etc.

Additional comments

Line 53: This makes it sound like an exact order of things that appears in all species, which is not the case. It also seems that authors are trying to describe the NCDS hypothesis here but loop back to it later.
Line 65: Wilkins is mentioned here when really it is what was described in line 53.
Line 78: This hypothesis is core to the paper. To help the readers understand the exact connections the hypothesis draws the authors should explain it in more detail rather than referring to the very recent original paper.
Line 79: From here on the introduction does not have a single citation. Where does all this information come from?
Line 96: How big are the enclosures? Free range could mean anything from a few hundred square meters to significantly more or less space. How long have animals been kept in these enclosures before? The parasite loads will add up over decades in these spaces that are consistently inhabited.
Line 118: I think this is supposed to say “treated”.
Line 127: So two different flotations methods were used? Were the used simultaneously for both countries or groups?
Line 131: There is a difference of overall parasite detection between the labs/countries. The authors state that this is irrelevant. Why? Could the authors expand?
Line 137: I think it would have been nice to either have the same methodology or still analyse the data in one group. This makes it somewhat difficult to compare. I think this data would have potentially benefitted from having a model run.
Line 174: The authors state: “According to PMD baselines, domestic pigs descend from parasite-susceptible ancestors, with susceptibility to parasites, which presumably trigger the domestication syndrome, being passed on/inherited through generations of progeny after the domestication process.” For the reader a more detailed explanation of how this all works from a mechanistic standpoint would be very helpful during the introduction.

Line 189: “none of the breeds included in our study had been historically subjected to methodical selection for parasite resistance.” – They may have been passively selected for that.
Line 197: “Since domestication, animals have been confined, at least to a certain extent, which prevents them from being exposed to the full spectrum of parasites that could infect/infest them in their natural environment.” – Personally I wouldn’t agree with either parts of that statement. Either way this sentence would benefit from references.
Line 200: The authors mention artificial selection here which would be in direct conflict of the mentioning of the PMD being more focussed on the proto-domesticates in the beginning.
Line 216: In light of this hypothesis I think it is crucial to carefully tease apart artificial selection from domestication.
Line 279: I am not sure what the authors are trying to say here, but it would also benefit from a citation.
Line 297: Just the correlation of presence of a parasite to the urban pigs does not convince me that wild boars wouldn’t have the same load if they were placed in the urban environment instead.
In general the discussion almost feels like a second introduction. I would suggest the authors edit it to a more concise length with a stronger overarching “story”. The fast jumping between species and various adjacent topics makes the discussion a little difficult to follow and makes the reader loose the connection to the previous parts of the paper and hypothesis.

·

Basic reporting

1. All the comments for correction/ modification have been highlighted on the attached reviewed manuscript file. Please modify accordingly.
2. Please redraft the Discussion in a very brief manner.
3. Write and cite references properly as per the format of the Journal.

Experimental design

All the comments for correction/ modification have been highlighted on the attached reviewed manuscript file. Please modify accordingly.

Validity of the findings

All the comments for correction/ modification have been highlighted on the attached reviewed manuscript file. Please modify accordingly.

---

## Round 0.2 · accepted · Accept

While the reviewers still disagree about the value of the manuscript, I think that it might be of interest to the community even with the listed caveats. Most editorial remarks have been mainly taken care of, although, as mentioned by reviewer 1, the paper might become more influential if more references were included in support of the main hypothesis. The fact that the studied groups live in similar conditions should be emphasized.

Reviewer 1 ·

Basic reporting

The 'parasite-mediated domestication' hypothesis is still very much unclear to me after reading both the original article it was published in and this manuscript. The underlying mechanism still eludes me. I think there is a lot of potential in this idea but I need to see it spelled out to evaluate what it is the authors are claiming. Not enough literature is cited in crucial points and huge parts of the introduction rely solely on the 2023 Skok original of the PM hypothesis.

Experimental design

The authors state: "wild populations (expected lower load/higher resistance) and domesticated populations (expected higher load/lower resistance)". I understand that this is supposed to lay the foundation as to how the hypothesis is tested but I do want to point out that simple exposure chances in smaller pens of domesticated pigs could also explain any potential differences. Even if a difference was found I wouldn't trust it to be due to domestication but would rather assume husbandry. To use this approach both groups would have to live in same conditions.

Validity of the findings

The results do not support the stated conclusions. The authors state: "Domestic pigs generally had a higher parasite load than their wild counterparts. The results of the present study, therefore, support one of the basic assumptions of PMD that domestic pigs could descend from parasite-susceptible wild ancestors." The results do not show that but actually show inconclusive results. Further the reader is not presented with any statistical numbers/data. We also wouldn't be able to deduct from this experiment that domestic pigs are more susceptible to parasites in general. They might just be more frequently exposed to them.

·

Basic reporting

No comment

Experimental design

No comment

Validity of the findings

No comment

Additional comments

All the corrections have been incorporated by the authors.